# CPF Induces GC2*_spd_* Cell Injury via ROS/AKT/Efcab6 Pathway

**DOI:** 10.3390/cells14130940

**Published:** 2025-06-20

**Authors:** Xuelian Zhang, Mengyang Zhang, Chunzhi Wang, Qingchuan Song, Haiyan Yang, Qi Tang, Qiaoling Zhao, Jing Wang, Chuanying Pan

**Affiliations:** 1Key Laboratory of Animal Genetics, Breeding and Reproduction of Shaanxi Province, College of Animal Science and Technology, Northwest A&F University, Yangling 712100, China; zhangxuelian199412@126.com (X.Z.); zhangmengyang0518@163.com (M.Z.); chunzhi_w122331@163.com (C.W.); songqingchuan@nwafu.edu.cn (Q.S.); yanghaiyanzsl@163.com (H.Y.); tangqi960@163.com (Q.T.); 2Jiangsu Key Laboratory of Sericultural Biology and Biotechnology, School of Biotechnology, Jiangsu University of Science and Technology, Zhenjiang 212100, China; qlzhao302@126.com; 3Henan Key Laboratory of Farm Animal Breeding and Nutritional Regulation, Henan Pig Breeding Engineering Research Centre, Institute of Animal Husbandry, Henan Academy of Agricultural Sciences, Zhengzhou 450002, China

**Keywords:** Chlorpyrifos, male infertility, RNA-Sequencing, PI3K-AKT, *Efcab6*

## Abstract

Chlorpyrifos (CPF) has been extensively utilized in recent decades due to its highly efficient insecticidal properties. However, the widespread use of pesticides has posed new challenges to male reproduction. This study aims to explore the potential molecular mechanisms of male reproductive decline induced by CPF. We employ flow cytometry, qRT-PCR, Western blot, RNA sequencing, and bioinformatics analysis to investigate the potential molecular mechanisms involved in CPF-induced male reproductive damage in GC2*_spd_* cells. Our results revealed that after 24 h of CPF treatment, the cell viability, cell cycle, apoptosis, and reactive oxygen species (ROS) accumulation of GC2*_spd_* cells were significantly affected in vitro. RNA sequencing analysis data indicated that a total of 626 genes were differentially expressed compared to the DMSO group, especially for *Efcab6*, *Nox3*, and *Cmpk2*. These differential genes were mainly enriched in signaling pathways such as PI3K-AKT and glutamine metabolism. In addition, further validation through qRT-PCR, Western blot, and experiments involving the inhibition of intracellular ROS generation with N-acetylcysteine collectively confirmed that CPF induces male reproductive damage through the ROS/AKT/Efcab6 pathway. These studies elucidate potential targets and molecular mechanisms underlying CPF-induced male infertility, providing a theoretical basis for the prevention of male reproductive damage caused by pesticide residues.

## 1. Introduction

In recent years, the decline in semen quality has attracted increasing attention globally. A study of semen quality changes in sperm donor candidates in Henan Province, China, showed significant decreases in volume, sperm concentration, total sperm count, progressive motility, and total motility between 2009 and 2019 [1]. The increasing use of pesticides has been one of the major contributing factors to the decline in semen quality [2]. Recently, it has been shown that annual spraying of organophosphorus pesticides resulted in a 6.253 unit decrease in sperm quality index from An Giang Province, Vietnam [3]. Chlorpyrifos (CPF) is one of the most widely used organophosphate pesticides in recent decades [4]. Organisms can be exposed to CPF through oral ingestion, inhalation, and dermal contact [5]. Even after the ban was enacted by the European Union in 2020, it continues to be used in some developing countries due to its highly efficient insecticidal properties [6]. The main insecticidal mechanism of CPF involves the inhibition of acetylcholinesterase (AChE) activity, leading to the induction of neurotoxicity [7]. However, AChE inhibition alone cannot explain all the symptoms of CPF toxicity, as CPF exposure can also affect other systems, such as inducing hematological, immune, and reproductive toxicities [8,9]. However, up to now, the targets and molecular mechanisms of CPF-induced reproductive toxicity are still unelucidated.

In rodents, during embryonic development, the expression of *Sry* triggers a battery of critical events. Primordial germ cells (PGCs) interact with Sertoli cells (SCs), Leydig cells (LCs), and myoid cells to complete sex determination, form seminiferous tubules, and constitute the embryonic testis [10,11]. Subsequently, PGCs undergo a period of mitotic division to prospermatogonia [12]. Once a threshold of prospermatogonia is reached, they enter a non-proliferative quiescent phase. Later, these prospermatogonia migrate to the spermatogonial stem cell (SSC) niche located on the periphery of the seminiferous tubules, where some prospermatogonia transition into SSCs [13,14]. Most SSCs remain undifferentiated until they begin to differentiate into A paired spermatogonia (Apr) or A aligned spermatogonia (Aal) after adolescence [15]. The majority of Aal spermatogonia transition to A1 differentiated spermatogonia without cell division. Subsequently, A1 differentiated spermatogonia generate A2, A3, A4, In, and B spermatogonia through cell divisions. These spermatogonia undergo a series of transformations and enter meiotic division, which leads to the formation of various types of spermatocytes and, ultimately, the development of round spermatids. These round spermatids then undergo spermatid morphogenesis to become elongated spermatids. This entire process takes approximately 35 days and is referred to as the cycle of the seminiferous epithelium. However, this process occurs every 8.6 days [16]. Previous studies have shown that CPF can impede sperm maturation in the epididymis [17]. To further investigate the potential molecular mechanism of CPF-induced reproductive impairment in male mice, this study selected GC2*_spd_* cells for experimentation. GC2*_spd_* cells represent a cell line that leans toward spermatocytes and possesses certain characteristics of sperm cells, providing better continuity with previous experiments.

In approximately 40–50% of male infertility cases, oxidative stress (OS) mechanisms are considered a significant cause of male fertility parameter disruptions [18]. Sperm, due to their limited antioxidant defense and DNA repair systems, are highly vulnerable to the deleterious influence of OS [19]. Research has indicated that low levels of reactive oxygen species (ROS) within spermatozoa are necessary for normal male reproduction. However, when OS occurs, an excess of ROS is generated, which can impact cellular components, nucleic acids, lipids, and proteins, leading to cell damage [20]. Notably, prior studies have shown that CPF exposure can increase OS in the mouse brain [21]. Additionally, our previous experiments have demonstrated that CPF can increase OS in the sperm of mouse cauda epididymidis. Therefore, this study aims to further validate these findings using GC2*_spd_* cells and explore the potential molecular mechanisms underlying CPF-induced male reproductive disorders.

Based on this, the experiment employed CPF treatment on the GC2*_spd_* cell line (a spermatocytes cell line, purchased from ATCC, Cat number CRL-2196) and utilized RNA sequencing to identify potential targets of CPF-induced male reproductive damage. This study aims to reveal the potential mechanisms of CPF-induced male reproductive toxicity and provide reference materials for the sustainable and healthy development of animal husbandry. The overarching goal of this research endeavor is to shed light on the intricate mechanisms underlying CPF-induced male reproductive toxicity, which aspires to furnish valuable reference materials that contribute to the sustainable and healthy development of animal husbandry practices.

## 2. Materials and Methods

### 2.1. Cell Culture and Treatments

The GC2*_spd_* cell line was purchased from ATCC: The Global Bioresource Center (https://www.atcc.org (accessed on 20 May 2025)) (CRL-2196, ATCC, Manassas, VA, USA). The medium contained DMEM/HIGH GLUCOSE (E600003, Sangon, Shanghai, China), fetal bovine serum (Z7186FBS-500, Zeta Life, New York, NY, USA), Penicillin-Streptomycin (15140122, Gibco, New York, NY, USA), GlutaMAX (35050061, Gibco, New York, NY, USA), AdvanceSTEM ES Qualified Non-Essential Amino Acids (100X) (SH30853.01, HyClone, Logan, UT, USA), and Sodium Pyruvate (2276850, Gibco, New York, NY, USA); their proportions were 86%, 10%, 1%, 1%, 1%, and 1%, respectively. Six-well plates were initially seeded with a cell density of 3.5 × 10^5^, and cells were cultured for 12 h and then treated with CPF. CPF was dissolved in Dimethyl sulfoxide (DMSO) (HY-Y0320, MedChemExpress, Monmouth Junction, NJ, USA). Then, 12.5 μmol/L CPF (CPF12.5) and 25 μmol/L CPF (CPF25) were finally selected to treat the GC2*_spd_* cells for 12 or 24 h based on published articles and pre-experiment results on GC2*_spd_* cells [17,22]. CPF came from Shanghai Aladdin Bio-Chem Technology Company, Ltd. (C109843, Shanghai, China). The same volume of DMSO was used as the control (DMSO).

### 2.2. The Detection of Cell Viability

According to the manufacturer’s instructions, the viability of GC2*_spd_* cells was assessed after treatment with 0 and 25 μmol/L CPF for 12 h using the Cell Counting Kit 8 (CCK8) (C0037, Beyotime Institute of Biotechnology, Shanghai, China). The specific experimental methods are as follows: 10 μL of CCK8 solution was added per 100 μL of cell culture medium; after incubation at 37 °C in darkness for 2~4 h, the absorbance of the cells was measured at 450 nm using a microplate reader.

### 2.3. The Detection of Cell Cycle

After treating GC2*_spd_* cells with 0 and 25 μmol/L CPF for 24 h, the floating dead cells in the supernatant were removed, and the live cells digested with trypsin were collected by centrifugation at 450× *g* for 5 min. Flow cytometry was used to detect each cell cycle. For detailed methods, refer to the article by Zhang et al. [23].

### 2.4. The Detection of Apoptosis

Apoptosis was detected using an apoptosis kit (Annexin V-FITC Apoptosis Detection Kit) (BA1150, Nanjing Biobox Biotech. Co., Ltd., Nanjing, China) on GC2*_spd_* cells treated with CPF (0 and 25 μmol/L) for 24 h according to the commercial instructions. More than 1 × 10^4^ cells were required in each tube. Subsequently, the treated cells were stained using Annexin-V-FLUOS and PI, respectively, and apoptosis was detected after incubation. The specific steps are described in Chen et al. [22].

In addition, the One-step TUNEL In Situ Apoptosis Kit (Green, Elab Fluor^®^ 488) (E-CK-A321, Elabscience Biotechnology Co., Ltd., Wuhan, China) was utilized for detecting late apoptosis cells in CPF-treated GC2*_spd_* cells. The specific method was described in the kit instructions. Firstly, GC2*_spd_* cells were fixed using 4% paraformaldehyde at room temperature for 20–30 min, and the fixed samples were immersed in PBS and rinsed 3 times for 5 min each time; subsequently, the cells were permeabilized using 0.2% Triton-100 at 37 °C for 10 min; the permeabilized samples were immersed in PBS and rinsed 3 times for 5 min each time; 100 μL of TdT Equilibration Buffer was added to each sample, equilibrated at 37 °C for 30 min. Prepare the labeling working solution and add an appropriate amount to each well, react at 37 °C for 60 min in the dark; rinse the labeled samples by immersing them in PBS for 3 times, for 5 min each time; add DAPI working solution and incubate for 5 min at room temperature, avoiding light; rinse the samples by adding PBS for 4 times, for 5 min each time. Finally, the stained cells were photographed and counted under a fluorescence microscope to calculate the late apoptosis rate of GC2*_spd_* cells after 25 μmol/L CPF treatment.

### 2.5. RNA Extraction and qRT-PCR

RNA in GC2*_spd_* cells treated with CPF (0, 12.5 and 25 μmol/L) was extracted using RNAiso Plus (9109, TAKARA, Beijing, China), chloroform, isopropanol, anhydrous ethanol, and enzyme-free water (R1600, Solarbio, Beijing, China). The RNA purity of samples was assessed by its OD260/280 value using a NanoDrop 1000 spectrophotometer (Thermo Fisher Scientific Inc., Wilmington, DE, USA). In addition, electrophoretic analysis was performed on 1% agarose gel to assess the RNA quality [24]. Then the cDNA of the GC2*_spd_* cells was obtained from RNA using Hifair^®^ II 1st Strand cDNA Synthesis SuperMix for qPCR (gDNA digester plus) (11123ES10, YEASEN, Shanghai, China) according to the manufacturer’s protocol. Briefly, genomic DNA is first removed, followed by reverse transcription of RNA to cDNA. cDNA obtained was stored in a −80 °C refrigerator. The gene expression levels in GC2*_spd_* cells were detected using qRT-PCR and performed using ChamQ SYBR qPCR Master Mix (Q311-02, vazyme, Nanjing, China). The primer sequences were designed by NCBI (https://blast.ncbi.nlm.nih.gov/Blast.cgi (accessed on 16 April 2023)) and are shown in Table 1, and the specific qRT-PCR reaction system and reaction procedures are shown in Appendix A.

### 2.6. Detection of Oxidative Stress Level

Relative ROS levels in GC2*_spd_* cells were measured using a ROS assay kit (S0033, Beyotime Institute of Biotechnology, Shanghai, China). The principle of this kit is that DCFH-DA itself is non-fluorescent, but intracellular ROS can oxidize non-fluorescent DCFH to produce fluorescent DCF, so the fluorescence of DCF can be detected to determine the level of ROS in the cell. Briefly, the GC2*_spd_* cells were collected, the fluorescent probe DCFH-DA in the ROS assay kit was added, and the cells were bathed in the dark at 37 °C for 30 min. The concentration of the DCFH-DA working solution is 10 μmol/L. Then, the cells washed three times using serum-free medium, and finally, the fluorescence intensity and absorbance values were measured using a microplate reader at 488 nm and 525 nm to calculate the intracellular ROS levels in GC2*_spd_* cells.

### 2.7. RNA-Seq and Data Analyze

The obtained RNA was sent to the company for sequencing, and the steps of Zhang et al. [23] were followed. Subsequently, the R package (https://www.omicstudio.cn/home (accessed on 16 April 2023)) was utilized for data analysis, including principal component analysis (PCA) and differentially expressed genes (DEGs), including GO, KEGG, Reactome enrichment, etc.

### 2.8. Western Blot (WB)

Proteins were collected from GC2*_spd_* cells after CPF treatment for 24 h using RIPA (PL001, Shaanxi Zhonghui Hecai Biological Medicine Technology Co., Ltd., Xi’an, China), then centrifuged at 12,000× *g* for 20 min in 4 °C to collect the supernatant, followed by the addition of loading buff (P1040, Solarbio, Beijing, China) and bathing at 100 °C for 20 min. Subsequently, the prepared protein samples were subjected to electrophoresis, transferred to PVDF membranes (ISEQ00010, Millipore, Burlington, MA, USA), sealed with skimmed milk at room temperature for 2 h, and incubated with primary antibodies at 4 °C for 12 h. Then, the secondary antibodies were incubated at room temperature for 2 h after being purged in TBST for 3 times. Protein expression was detected using Ultra-sensitive ECL Reagent (DY30208, DIYIBIO, Shanghai, China) and Bio-Rad Chemidoc. Details of the primary and secondary antibodies are shown in Appendix A.

### 2.9. The Prediction of Molecular Docking

The prediction of molecular docking was performed in the 3D protein structures of candidate genes and CPF using HOME for Researchers (https://www.dockeasy.cn/DockCompound (accessed on 16 April 2023)). When the binding energy between the ligand small molecule and the receptor protein is less than −4.25 kcal/mol, it indicates that there is some binding activity between the two; when the binding energy is less than −5.0 kcal/mol, it can be regarded as good binding activity; when the binding energy is less than −7.0 kcal/mol, it suggests that there is strong binding activity.

### 2.10. SC79 and CPF Co-Treatment

SC79 was purchased from MedChemExpress (HY-18749, Monmouth Junction, NJ, USA). GC2*_spd_* cells were passaged to 6-well (3.5 × 10^5^) plates and cultured for 12 h. After 12 h, 0, 5, and 10 μmol/L SC79 was added into the medium for 24 h, and SC79 was dissolved in DMSO. Then, 5 μmol/L SC79 was selected and added into culture medium for pre-protecting the adhered GC2*_spd_* cells. After 1 h of pre-protection, CPF (0, 12.5, and 25 μmol/L) was added into the medium to co-treat for 24 h. The cells were named DMSO + DMSO, DMSO + CPF12.5, DMSO + CPF25, SC79 + DMSO, SC79 + CPF12.5, and SC79 + CPF25, respectively. Subsequently, the co-treated GC2*_spd_* cells were assayed for cellular morphology and CCK8.

### 2.11. N-Acetylcysteine (NAC) and CPF Co-Treatment

NAC scavenges intracellular ROS by enhancing the intracellular cysteine pool, increasing glutathione (GSH) levels and enhancing the activity of antioxidant enzymes (glutathione peroxidase, thioredoxin) [25]. NAC was purchased from MedChemExpress (HY-B0215, Monmouth Junction, NJ, USA). GC2*_spd_* cells were passaged to 6-well (3.5 × 10^5^) or 96-well (2 × 10^4^) plates and cultured for 12 h. After 12 h, NAC was dissolved in H_2_O and 0 and 5 mmol/L NAC were added for pre-protection. The choice of NAC dose was screened according to Zhang and Wei et al. and no toxic effect of 5 mmol/L NAC on GC2*_spd_* cells was detected in pre-tests [26,27]. After 1 h of pre-protection, CPF (0, 12.5, and 25 μmol/L) was added into the medium to co-treat for 24 h. The cells were named H_2_O + DMSO, H_2_O + CPF12.5, H_2_O + CPF25, NAC + DMSO, NAC + CPF12.5, and NAC + CPF25, respectively. Subsequently, the co-treated GC2*_spd_* cells were assayed for cellular morphology, CCK8, ROS, and gene expression.

### 2.12. Statistical Analysis

SPSS 23.0 software was used for statistical analysis of the data. Independent samples *t*-test (*t*-test) or one-way analysis of variance (ANOVA) was used to analyze the differences between all groups. When *p* < 0.05 or< 0.01, these differences were considered significant or extremely significant, respectively. All data are expressed as mean ± standard error of the mean (SEM). And all biological replicates in the experiments were more than or equal to 3 (*n* ≥ 3).

## 3. Results

### 3.1. The Morphology and Viability of GC2_spd_ Cells After CPF Treatment

Preliminary research findings have indicated that CPF has male reproductive toxicity in vivo [17]. In order to further investigate the potential mechanisms of male infertility induced by CPF in mice, the GC2*_spd_* cell line, which is biased towards spermatocytes, was selected for in vitro validation. Firstly, CPF was added to the culture medium of GC2*_spd_* cells for 12 h and 24 h of in vitro experiments. After treatment with CPF at 12.5 and 25 μmol/L for 12 h, there was a significant reduction in adherent GC2*_spd_* cells (Figure 1A); counting the GC2*_spd_* cells after 24 h of CPF treatment, it was found that the CPF12.5 and CPF25 treatment groups exhibited approximately a 20% decrease in cell number (Figure 1B). Additionally, CCK8 results showed that exposure to 25 μmol/L CPF for 12 h led to a significant decrease in cell viability of GC2*_spd_* cells compared to the control group (*p* = 0.032) (Figure 1C).

### 3.2. Cell Cycle Arrest of GC2_spd_ Cells After CPF Treatment

After CPF treatment, the reduction in adherent GC2*_spd_* cells may be attributed to factors such as decreased cell proliferation or increased cell apoptosis. Based on this, cell cycle analysis was performed on GC2*_spd_* cells after 24 h of CPF treatment. Firstly, flow cytometry showed that in the DMSO group, the proportions of cells in the G1, G2, and S phases were 83.97 ± 0.60%, 12.76 ± 0.85%, and 3.27 ± 0.44%, respectively; after CPF treatment, the proportions of cells in the G1, G2, and S phases were 70.95 ± 0.26%, 16.82 ± 0.68%, and 12.24 ± 0.56%, respectively (Figure 2A,B). After 24 h of CPF treatment, there was a significant decrease in G1 phase cells (*p* = 0.000001), and a significant increase in S phase (*p* = 0.000015) and G2 phase (*p* = 0.009669) cells, indicating that GC2*_spd_* cells were arrested in the S and G2 phases (Figure 2C).

Subsequently, the expression of cell cycle-related genes was examined. After 24 h of CPF treatment, the CPF25 group showed significant upregulation of *P53* (*p* = 0.014) and *P21* (*p* = 0.030) (Figure 2D,E), while the expression of cell cycle-related genes *Ccna* (*p* = 0.002), *Ccnd* (*p* = 0.041), *Ccnb* (*p* = 0.048), and *Ccne* (*p* = 0.003) was significantly downregulated (Figure 2F–I). These genes collectively regulate GC2*_spd_* cells, leading to cell cycle arrest in the S and G2 phases after 24 h of CPF treatment.

### 3.3. Apoptosis of GC2_spd_ Cells After CPF Treatment

An increase in apoptosis is also one of the important reasons for the decrease in the number of adherent cells. Subsequently, the apoptosis of CPF-treated GC2*_spd_* cells was investigated. First, the TUNEL assay was used to detect GC2*_spd_* cells treated with 25 μmol/L of CPF for 24 h. The results showed that after 24 h of CPF treatment, the proportion of TUNEL-positive cells in the CPF25-treated group was significantly higher than that in the DMSO group (*p* = 0.016) (Figure 3A,B). The proportion of TUNEL-positive cells in the DMSO group was 0.8424 ± 0.10372, while in the CPF25-treated group, it was 3.1959 ± 0.50442.

Flow cytometry analysis showed that after 24 h of CPF treatment, the proportion of early apoptosis, late apoptosis, and necrotic cells in the CPF25-treated group was significantly higher than that in the DMSO group (*p* = 0.009, *p* = 0.040, and *p* = 0.005), and the proportion of live cells was significantly lower than the control group (*p* = 0.000241) (Figure 3C–F). The proportions of live cells, early apoptotic cells, late apoptotic cells, and necrotic cells in the DMSO group were 90.6500 ± 0.43684, 2.10000 ± 0.43970, 5.2250 ± 0.22867, and 2.0250 ± 0.29262, respectively; while in the CPF25 group, they were 81.6000 ± 0.98489, 4.70000 ± 0.40415, 9.0667 ± 0.86859, and 4.6333 ± 0.50442, respectively.

Apoptosis-related gene detection showed that after 24 h of CPF treatment, there was an upward trend in the expression of apoptosis-related genes (*Caspase3* and *Caspase9*), but it was not statistically significant (Figure 3G,H), suggesting the possible presence of post-transcriptional regulation.

### 3.4. The Oxidative Stress Level of GC2_spd_ Cells After CPF Treatment

It is known that CPF can cause increased oxidative stress levels in the mouse cauda epididymidis. Therefore, further investigation was carried out to measure the oxidative stress levels in GC2*_spd_* cells after 24 h of CPF treatment. The results showed that after 24 h of CPF treatment, the ROS levels significantly increased in the 25 μmol/L treatment group (*p* = 0.025) (Figure 4A), which is consistent with the in vivo results. The elevation of ROS levels is associated with mitochondrial dysfunction. Subsequently, the expression of mitochondrial marker genes was examined, and it was found that there was no significant change in *Vdac1* expression, but the expression level of *Cox4* significantly increased (*p* = 0.010) (Figure 4B,C). This indicates that mitochondrial dysfunction occurs after CPF treatment, leading to an increase in oxidative stress levels.

### 3.5. RNA-Seq of GC2_spd_ Cells After CPF Treatment

Based on the above results, to investigate the potential mechanisms underlying CPF-induced male reproductive damage, RNA sequencing was performed on GC2*_spd_* cells treated with CPF for 24 h, and the sequencing data was analyzed. The violin plots show that the quality of the sequencing data met the criteria for further analysis (Figure 5A). Results from PCA and heatmaps indicated good consistency within the group, confirming the suitability of the samples for subsequent analysis (Figure 5B,C). Subsequently, statistical analysis was conducted on the differentially expressed genes. The results showed that after 24 h of CPF treatment at 25 μmol/L, a total of 626 genes were differentially expressed compared to the DMSO group, with 267 upregulated and 359 downregulated (Figure 5D). Further analysis of the differentially expressed genes revealed significant upregulation of the male reproductive-related gene *EF-hand calcium binding domain 6* (*Efcab6*), significant upregulation of the oxidative stress-related gene *NADPH oxidase 3* (*Nox3*), and significant downregulation of the mitochondrial function-related gene *cytidine/uridine monophosphate kinase 2* (*Cmpk2*). These genes will be the focus of further study. Additionally, GO enrichment analysis of the differentially expressed genes showed that compared to the DMSO group, the CPF25 group exhibited significant enrichment in pathways such as cellular immunity, inflammation response, and protein synthesis (Figure 5E). The KEGG enrichment analysis of the differentially expressed genes identified the top 20 pathways including the TNF and Jak-STAT pathways that were significantly enriched in the CPF25 group compared to the DMSO group (Figure 5F). In addition, the top 50 pathways involved TNF, Jak-STAT, PI3K-AKT, and glutamine metabolism (Figure 5F). Lastly, analysis using Reactome revealed that differentially expressed genes were mainly enriched in cellular immunity, inflammation, and the Jak-STAT pathway (Figure 5G).

### 3.6. The Gene and Protein Expression of GC2_spd_ Cells After CPF Treatment

Based on the aforementioned sequencing and bioinformatics analysis results, male reproductive-related genes (*Efcab6*), oxidative stress-related genes (*Nox3* and *Cmpk2*), immune-related genes (*Ifi44l*, *Zbp1*), and Jak-STAT pathway-related genes (*Jak2*, *Jak3*, *Jak1*, *Stat1*, and *Stat2*) were selected and validated. After 24 h of CPF treatment in GC2*_spd_* cells, the expression of the male reproductive-related gene (*Efcab6*) was significantly upregulated (*p* = 0.005) (Figure 6A), which can inhibit the expression of the androgen receptor (*AR*). The expression of the oxidative stress-related gene (*Nox3*) significantly increased (*p* = 0.001) (Figure 6B), while *Cmpk2* expression significantly decreased (*p* = 0.016) (Figure 6F). The expression of *Ifi44l* and *Zbp1* was significantly downregulated (*p* = 0.000042, *p* = 0.003) (Figure 6G,H). The expression of Jak-STAT pathway-related genes also demonstrated significant changes. *Jak2* expression was significantly upregulated (*p* = 0.005), while *Stat1* and *Stat2* expression were significantly downregulated (*p* = 0.025, *p* = 0.020). *Jak1* and *Jak3* showed an upward trend in expression but were not statistically significant (*p* > 0.05) (Figure 6C–E,I–K). These results are consistent with the RNA sequencing analysis.

Subsequently, based on the sequencing results, related signaling pathways were also screened. In Figure 4A, the results showed a significant increase in ROS levels in GC2*_spd_* cells after CPF treatment, and in Figure 5F, the results indicated that differentially expressed genes were enriched in the glutamine metabolism pathway. Therefore, it was speculated that CPF may induce ferroptosis in GC2*_spd_* cells. Then, ferroptosis-related proteins were detected in GC2*_spd_* cells treated with CPF for 24 h. The results showed that the expression of ferroptosis-related protein ferritin heavy polypeptide 1 (FTH1) did not change significantly (Appendix A), while transferrin receptor 71 (CD71) expression was significantly reduced (Appendix A). These results suggest that CPF may not affect the survival of GC2*_spd_* cells through inducing iron death.

In addition, existing studies have reported that the accumulation of a large amount of intracellular ROS can inhibit the activity of the PI3K-AKT signaling pathway. Combining the results of mouse GC2*_spd_* cell sequencing analysis and sequencing analysis in pig ST cells [23], it is speculated that CPF may induce male reproductive damage through the ROS/PI3K/AKT pathway. Therefore, the protein expression of the PI3K-AKT signaling pathway was further verified. The results showed that after 24 h of CPF treatment, the expression of AKT in GC2*_spd_* cells was significantly decreased, and the downstream effector protein p-AKT expression was significantly lower than in the control group, indicating inhibition of the PI3K-AKT signaling pathway (Figure 6L). After that, SC79 was utilized to activate the phosphorylation of AKT in GC2*_spd_* cells. The optimal concentration of SC79 was first examined in GC2*_spd_* cells, and the results showed that 5 μm/L SC79 did not have a damaging effect on GC2*_spd_* cells; 5 μm/L SC79 was used for subsequent experiments (Appendix A). However, the addition of SC79 did not rescue the decrease in the number and viability of GC2*_spd_* cells caused by CPF (Appendix A). The above experiments indicated that CPF reduced p-AKT expression by decreasing the total AKT level in GC2*_spd_* cells and thus did not affect the phosphorylation process of AKT.

Next, the prediction of molecular docking was performed for genes with large fold changes after CPF treatment, such as *Efcab6*, *Nox3*, and *Zbp1*. Firstly, the 3D structures of the corresponding proteins for these genes were searched in the RCSB PDB database (https://www.rcsb.org (accessed on 16 April 2023)), where NOX3 had no available structure resolution, the PDB ID for EFCAB6 was 1WLZ, and the PDB ID for ZBP1 was 2HEO. In addition, the prediction of molecular docking was also performed for AR (1GS4), the downstream target protein of EFCAB6. According to the simulation results, CPF showed good binding affinity with AR with a binding energy of −6.369 and the formation of hydrogen bonding (Figure 6M); CPF also exhibited good binding affinity with EFCAB6 with a binding energy of −5.364, although no hydrogen bonding was formed (Figure 6N); whereas CPF had poor binding affinity with ZBP1 with a binding energy of −1.262 and no binding occurred (Figure 6O). These results suggest that CPF may directly target the structural domain of EFCAB6 or AR to regulate their expressions, thus reducing male reproductive fertility.

### 3.7. The Morphology of GC2_spd_ Cells After NAC and CPF Co-Treatment

This experiment utilized NAC to clear ROS in GC2*_spd_* cells, thereby reverse validating that CPF can damage GC2*_spd_* cells through ROS. After the co-treatment of NAC and CPF, live cells were significantly reduced after treatment with H_2_O + CPF12.5 and H_2_O + CPF25 for 24 h. However, the addition of NAC could restore this reduction (NAC + CPF12.5 and NAC + CPF25) (Figure 7A,A′). The above cells were counted, and it was found that 5 mmol/L of NAC could partially restore the reduction in cell number induced by CPF12.5 (Figure 7C); at the same time, the same dose of NAC could also partially restore the reduction in cell number induced by CPF25 (Figure 7C′). The above results indicated that NAC could restore CPF-induced damage in GC2*_spd_* cells after inhibiting intracellular ROS generation.

To further investigate the restorative effect of NAC on CPF damage, CCK8 assay was subsequently performed on GC2*_spd_* cells after co-treatment with NAC and CPF. The results showed that cell viability was significantly reduced (*p* = 8.2693 × 10^−10^) after H_2_O + CPF25 treatment of GC2*_spd_* cells for 24 h (Figure 7B′). After adding NAC for 1 h of pre-protection followed by co-treatment with CPF (NAC + CPF12.5 and NAC + CPF25), cell viability was significantly higher in the NAC + CPF12.5 group than that in the H_2_O + CPF12.5 (*p* = 0.006) (Figure 7B); cell viability in the NAC + CPF25 group was lower than that in the H_2_O + DMSO group, but significantly higher than that in the H_2_O + CPF25 group (*p* = 0.000005) (Figure 7B′). The above results indicated that the CPF-induced reduction in GC2*_spd_* cell viability could be restored after NAC inhibited intracellular ROS generation.

### 3.8. The ROS Levels of GC2_spd_ Cells After NAC and CPF Co-Treatment

Subsequently, ROS assay was performed on GC2*_spd_* cells after co-treatment with NAC and CPF, and the results showed that ROS levels in GC2*_spd_* cells were significantly increased after 24 h of treatment with H_2_O + CPF12.5 (*p* = 0.043) (Figure 7D); ROS levels in GC2*_spd_* cells were further increased in the H_2_O + CPF25-treated group, which was significantly higher than that in the H_2_O + DMSO group (*p* = 0.002) (Figure 7D′). However, after adding NAC for 1 h of pre-protection followed by CPF for treatment (NAC + CPF12.5 and NAC + CPF25), the intracellular ROS levels in GC2*_spd_* cells in the NAC + CPF12.5-treated group were lower than those in the H_2_O + DMSO group and lower than those in the H_2_O + CPF12.5 group, and the accumulation of ROS in the GC2*_spd_* cells was reduced by about 2-fold (*p* = 0.001) (Figure 7D); the intracellular ROS levels in GC2*_spd_* cells in the NAC + CPF25-treated group were also significantly lower than those in the H_2_O + DMSO and H_2_O + CPF25 groups, and the ROS accumulation was reduced by about 3-fold (*p* = 0.004) (Figure 7D’). The above results indicated that NAC inhibition of intracellular ROS generation restored CPF-induced ROS accumulation in GC2*_spd_*.

### 3.9. Apoptosis of GC2_spd_ Cells After NAC and CPF Co-Treatment

The expression of the above apoptosis-related genes was immediately explored. A 12.5 μmol/L CPF (H_2_O + CPF12.5) treatment of GC2*_spd_* cells for 24 h resulted in a significant increase in *Bax* expression in the GC2*_spd_* cells (*p* = 0.021); *Bax* expression in NAC + CPF12.5 was higher than that in the H_2_O + DMSO group, but lower than that in the H_2_O + CPF12.5 group (Figure 7E). In addition, the qRT-PCR results showed that *Bax* expression was significantly upregulated in the H_2_O + CPF25 group (*p* = 0.010), and in the NAC + CPF25, the expression of *Bax* also tended to be lower than that in H_2_O + CPF25 group (Figure 7E′). All these results indicated that NAC could partially restore the upregulation of apoptosis-related gene expression in GC2*_spd_* cells induced by 12.5 μmol/L and 25 μmol/L CPF.

### 3.10. Cell Cycle Arrest of GC2_spd_ Cells After NAC and CPF Co-Treatment

Subsequently, NAC was utilized to restore the alterations in proliferation and cell cycle-related gene expression in GC2*_spd_* cells induced by CPF. The results showed that the expression of *P21* was significantly elevated in GC2*_spd_* cells after 24 h of treatment with H_2_O + CPF12.5 (*p* = 0.001); the expression of *Pcna*, *Ccnd*, *Ccne* (*p* = 0.027), *Ccna* (*p* = 0.005), and *Ccnb* (*p* = 0.007) were lower than that in the control group (H_2_O + DMSO) (Figure 7F–K). However, the addition of NAC for 1 h of pre-protection followed by the addition of 12.5 μmol/L CPF for treatment (NAC + CPF12.5) showed that the expression of *P21* in the cells, although higher than that in the H_2_O + DMSO group, was lower than that in the H_2_O + CPF12.5; moreover, the expression of *Pcna*, *Ccnd*, and *Ccna* in GC2*_spd_* cells in the NAC + CPF12.5-treated group was lower than that in the control group (H_2_O + DMSO), but higher than that in the H_2_O + CPF12.5 group (Figure 7G–I). In the 25 μmol/L CPF (H_2_O + CPF25) treatment group, the results demonstrated that *P21* expression was significantly elevated in GC2*_spd_* cells after 24 h (*p* = 0.002); the expression of *Pcna*, *Ccnd* (*p* = 0.030), *Ccne* (*p* = 0.002), *Ccna* (*p* = 0.001), and *Ccnb* (*p* = 0.001) was lower than that in the control group (H_2_O + DMSO) (Figure 7F′–K′), which is consistent with the damage induced by 12.5 μmol/L CPF. After adding NAC for 1 h of pre-protection and then adding 25 μmol/L CPF for treatment (NAC + CPF25), the expression of *P21* in the cells was lower than that of H_2_O + CPF25 although higher than that of the H_2_O + DMSO group (Figure 7G′); moreover, the expression of *Ccnd*, *Ccne*, *Ccna*, and *Ccnb* of the GC2*_spd_* cells in the NAC + CPF25-treated group were lower than those in the control group (H_2_O + DMSO) but higher than those in the 25 μmol/L CPF-treated group (H_2_O + CPF25) (Figure 7H′–K′). All of the above results indicated that NAC inhibition of intracellular ROS generation partially restored the CPF-induced alteration of cell cycle-related gene expression in GC2*_spd_*.

### 3.11. The Expression of Efcab6 in GC2_spd_ Cells After NAC and CPF Co-Treatment

Previous results showed that CPF might have a direct targeting effect with *Efcab6*, so the expression of the potential target gene *Efcab6* in GC2*_spd_* cells after co-treatment with NAC and CPF was examined. The qRT-PCR results indicated that the expression of *Efcab6* in GC2*_spd_* cells was significantly elevated after treatment with 12.5 μmol/L CPF (H_2_O + CPF12.5) and 25 μmol/L CPF (H_2_O + CPF25) for 24 h (*p* = 0.002; *p* = 0.001) (Figure 7L,L′). But concerning the addition of NAC to pre-protect the cells for 1 h followed by the addition of CPF for co-treatment, the expression of *Efcab6* was higher than that in the H_2_O + DMSO group, but notably lower than that in the H_2_O + CPF12.5 and H_2_O + CPF25 groups (*p* = 0.013; *p* = 0.054) (Figure 7L,L′). These findings indicated that NAC partially restored the upregulation of potential target gene *Efcab6* expression in GC2*_spd_* cells induced by CPF.

## 4. Discussion

The advent of industrialization has undeniably positioned pesticides as vital components of agriculture and everyday life. However, the widespread use of pesticides also has brought about numerous adverse consequences. Pesticide residues can accumulate in animals through biomagnification and have an impact on the reproductive system, especially causing significant damage to the male reproductive system. Numerous studies have demonstrated that pesticide residues can affect male reproduction. Preliminary experiments have shown that CPF causes reproductive damage in male mice by reducing the expression of genes related to steroid synthesis, damaging mitochondrial function, increasing sperm oxidative stress, and reducing sperm motility and density, ultimately leading to male infertility [17]. However, the potential targets of CPF-induced male reproductive damage have not yet been identified. In light of this consideration, the primary objective of this experiment is to unveil the intricate mechanisms underpinning CPF-induced male reproductive toxicity, providing reference materials for the prevention and treatment of male reproductive disorders.

After CPF treatment of GC2*_spd_* cells, there was a decrease in adherent cells, cell cycle arrest, a significant increase in apoptotic cells, and a significant elevation of ROS levels, aligning with the previous in vivo experiments in mice. In-depth RNA sequencing analysis showed that after 24 h of treatment with 25 μmol/L CPF, several key pathways were enriched, including glutamine metabolism, Jak-STAT signaling, and PI3K-AKT pathways. Previous studies have shown that the Jak-STAT pathway is essential for self-renewal of male SSCs [28]. And in embryonic stem cells (ESs), Jak-STAT signaling is necessary for maintaining ESs [29]. Singh et al. showed that Jak-STAT signaling was also shown to be counteracted by Ras-Raf-MAPK, and that the two signals can subsequently converge on a number of downstream targets to co-regulate cell fate [29]. The RNA sequencing results suggest that in addition to its classic regulatory roles, the Jak-STAT pathway may be involved in CPF-induced damage in GC2*_spd_* cells. However, it is crucial to note that further research is warranted to delve deeper into the specific mechanisms.

Ferroptosis is a relatively novel form of regulated cell death that occurs when there is a significant increase in intracellular iron levels. This leads to the buildup of lipid peroxides within the cell membrane, which subsequently breaks down into aldehydes, reactive oxygen species, and other active derivatives, causing damage to cellular proteins, lipids, and nucleic acids, ultimately resulting in cell death [30]. Previous studies have shown that ferroptosis has negative effects on male reproduction [31], and recent research has shown that ferroptosis is involved in the death of sperm caused by bilateral varicocele [32]. In recent years, there has been a growing interest in exploring the adverse effects of ferroptosis in studies focused on male reproductive toxicity. Wang and Zeng et al. have shown that in cadmium-treated models, ferroptosis can inhibit testicular development, impair spermatogenesis, and affect testosterone synthesis, consequently reducing male fertility [33,34]. Additionally, Phthalic acid esters (PAEs) can induce ferroptosis in the testes, damage the blood–testis barrier (BTB), and inhibit male fertility [35]. Based on this, our experiment employed KEGG analysis, which revealed alterations in the glutamine metabolism pathway. When combined with the significant increase in oxidative stress levels in mouse sperm and GC2*_spd_* cells after CPF treatment, we speculated that CPF may induce male reproductive damage through the ferroptosis pathway. However, a closer examination of ferroptosis marker proteins showed that after 24 h of CPF treatment in GC2*_spd_* cells, there was no significant change in FTH1 expression, but CD71 expression was significantly reduced, indicating that CPF does not affect the viability of GC2*_spd_* cells through the induction of ferroptosis. It is important to note that CD71 expression is dependent on p-AKT, so the reduction in CD71 expression may be attributed to a decline in p-AKT levels [36,37].

The PI3K/AKT pathway is crucial for regulating various cellular processes and serves as an important signal transduction pathway in cell growth, energy conversion, cell cycle, apoptosis, and autophagy [38]. It also holds importance in male reproduction [39,40]. KEGG analysis and Western blot results showed that after CPF treatment, the levels of ROS were significantly elevated in GC2*_spd_* cells. Simultaneously, the activity of the PI3K/AKT pathway was inhibited. Previous studies have shown that in colorectal cancer research, Delicaflavone can induce the accumulation of ROS, suppress the PI3K/AKT/mTOR and Ras/MEK/Erk signaling cascades, inhibit the proliferation of colorectal cancer cells, and promote apoptosis [41]. Zhu et al. also demonstrated that avibactam can induce apoptosis and autophagy in TM3 cells by accumulating ROS and inhibiting the PI3K/AKT signaling pathway [42]. Additionally, related studies have shown that CPF can inhibit the PI3K/AKT signaling pathway, leading to apoptosis in brain and liver tissues [43,44]. Moreover, high levels of ROS can inhibit the activation of AKT through PI3K [45]. Therefore, CPF may induce apoptosis in GC2*_spd_* cells and reduce male fertility through the ROS/AKT pathway.

Numerous studies have established the endocrine-disrupting function of CPF in males, resulting in a decrease in testosterone synthesis and an increase in follicle-stimulating hormone (FSH) and luteinizing hormone (LH) levels [46]. The action of androgens in target cells is mediated by the androgen receptor (AR). Hazarika et al. harnessed techniques such as the prediction of molecular docking and molecular dynamics to demonstrate the molecular interactions between CPF and its degradation products with AR, identifying them as potent disruptors of androgen activity [47]. As early as 2003, researchers determined that *Efcab6* (also called *DJBP*) can bind to the COOH-terminal region of *DJ-1*. It also can bind to the DNA-binding domain of *AR*, both in vitro and in vivo. This interaction leads to the recruitment of histone deacetylase (HDAC) complexes, including HDAC1 and mSin1, ultimately culminating in the inhibition of AR activation [48]. Based on sequencing data, the *Efcab6* gene exhibited differential expression in GC2*_spd_* cells before and after CPF treatment, with particularly high expression after CPF treatment. We postulated that CPF may regulate the expression of the *Efcab6* gene to inhibit AR activity, in turn, diminishing male fertility. In line with this, the prediction of molecular docking results also supports our speculation. Furthermore, inhibiting intracellular ROS with NAC also resulted in decreased expression of the *Efcab6* gene, suggesting that CPF can induce apoptosis and reduce male fertility in GC2*_spd_* cells through the ROS/AKT/Efcab6 pathway.

Comparative analysis of mouse and pig transcriptome sequencing data revealed that differentially expressed genes were enriched in pathways such as PI3K-AKT, Jak-STAT, and inflammation [23]. Subsequent experiments also confirmed the toxic effects of CPF on mouse GC2*_spd_* and ST cells through the ROS/PI3K/AKT pathway. This implies a certain degree of conservation in the potential molecular mechanisms underlying CPF-induced male reproductive damage across different species. However, due to differences in species and cell types, the action mode of CPF and downstream effect genes may vary. For example, in GC2*_spd_* cells, CPF reduces AKT expression and subsequently decreases p-AKT. In contrast, in ST cells, CPF directly affects the phosphorylation process of AKT, reducing p-AKT expression. Additionally, in GC2*_spd_* cells, the expression of the male reproductive-related gene *Efcab6* is significantly upregulated, whereas in ST cells, the expression of the male reproductive-related gene *Hsd3b1* is significantly downregulated. Moreover, these genes share similar effects. Previous studies have shown that high expression of *Efcab6* in cells can inhibit transactivation of *AR*, thereby suppressing *AR* expression and ultimately causing male infertility [48]. Conversely, decreased expression of *Hsd3b1* can lead to a reduction in androgen synthesis, such as testosterone, ultimately causing male reproductive damage [49].

In this study, we examined in detail the potential mechanisms of male injury induced by CPF. However, unfortunately, we have not yet performed recovery experiments in vivo using NAC. In subsequent studies, we will aim to further investigate the restorative effects of NAC on CPF-induced male reproductive injury in vivo and to find natural compounds to replace the function of NAC from a food-borne perspective.

## 5. Conclusions

In summary, CPF can affect the cell viability, cell cycle, apoptosis, and ROS levels in GC2*_spd_* cells. And RNA sequencing illuminated that CPF can regulate the survival of GC2*_spd_* cells through the PI3K-AKT pathway. Rescue experiments using NAC confirm that CPF can activate ROS accumulation in GC2*_spd_* cells, thereby inhibiting the PI3K-AKT signaling pathway and activating *Efcab6* expression, leading to a decrease in cell survival.

## Figures and Tables

**Figure 1 cells-14-00940-f001:**
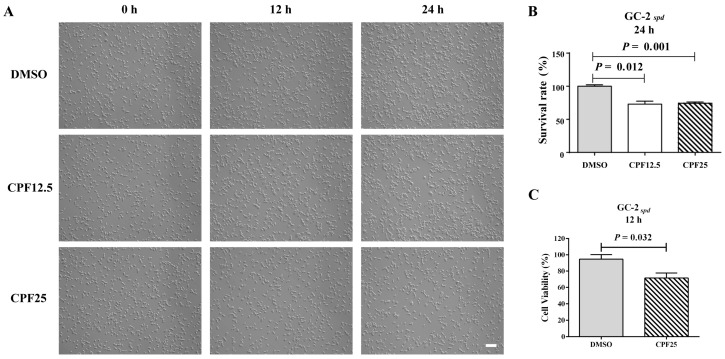
Status of GC2*_spd_* cells after CPF treatment for 24 h. (**A**) Phenotypes of GC2*_spd_* cells; (**B**) cell number of GC2*_spd_* cells; (**C**) cell viability of GC2*_spd_* cells. Scale bar indicates 100 μm.

**Figure 2 cells-14-00940-f002:**
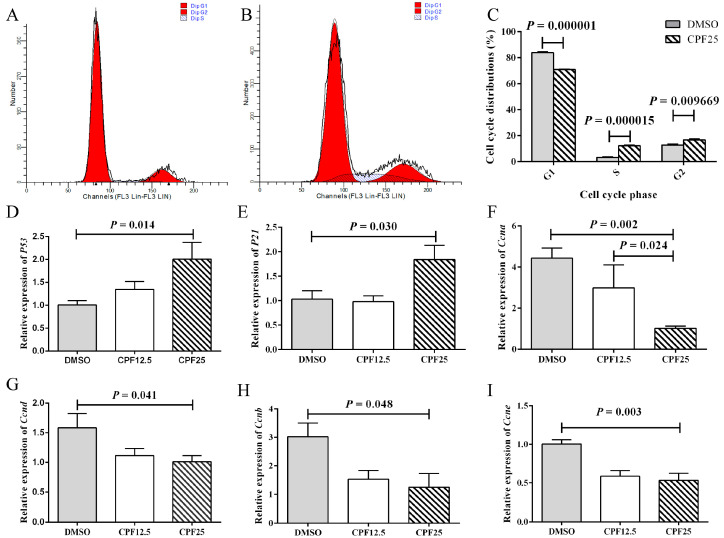
Cell cycle arrest of GC2*_spd_* cells after CPF treatment for 24 h. (**A**) Cell distributions of vehicle (DMSO) and CPF (25 μmol/L) group; (**B**) cell distributions of CPF (25 μmol/L) group; (**C**) statistics on cell cycle distribution; (**D**) mRNA expression of *P53*; (**E**) mRNA expression of *P21*; (**F**) mRNA expression of *Ccna*; (**G**) mRNA expression of *Ccnd*; (**H**) mRNA expression of *Ccnb*; (**I**) mRNA expression of *Ccne*.

**Figure 3 cells-14-00940-f003:**
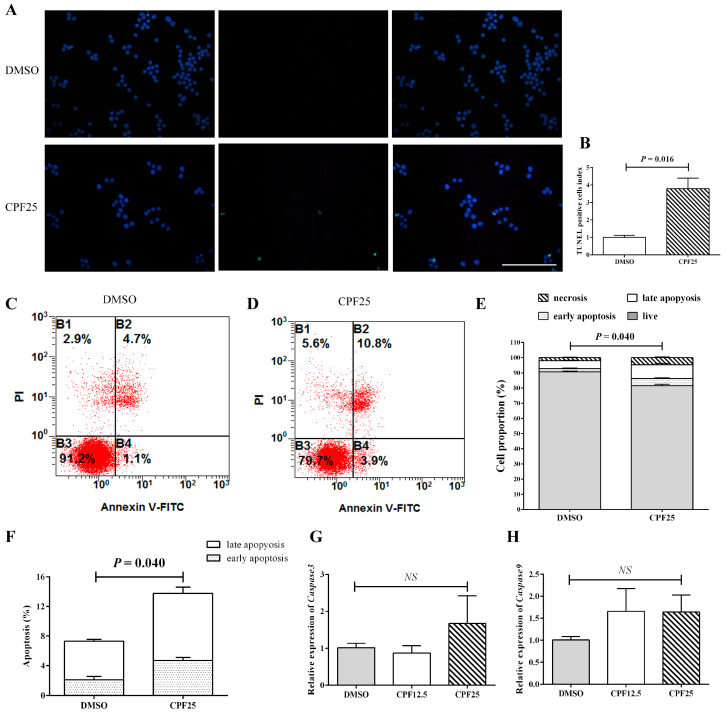
Detection of apoptosis in GC2*_spd_* cells after CPF treatment for 24 h. (**A**) TUNEL staining in GC2*_spd_* cells; (**B**) TUNEL cell positivity rate in GC2*_spd_* cells; (**C**,**D**) cell distributions of vehicle (DMSO) and CPF (25 μmol/L) group; (**E**,**F**) statistics of apoptotic cells in GC2*_spd_* cells; (**G**) mRNA expression of *Caspase3*; (**H**) mRNA expression of *Caspase9*. Blue fluorescence represents cells stained with DAPI, and green fluorescence indicates apoptotic cells. B1: membrane rupture cells; B2: represents late apoptotic cells and necrotic cells; B3: live cells; B4: early apoptotic cells. Scale bar indicates 100 μm. “*NS*” represents no significant difference between each group.

**Figure 4 cells-14-00940-f004:**
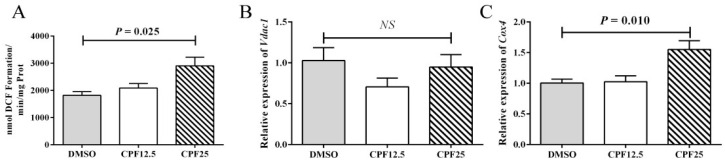
Mitochondrial homeostasis assay in GC2*_spd_* cells after 24 h of CPF treatment. (**A**) ROS levels in GC2*_spd_* cells; (**B**) expression of *Vdac1*; (**C**) expression of *Cox4*. “*NS*” represents no significant difference between each group.

**Figure 5 cells-14-00940-f005:**
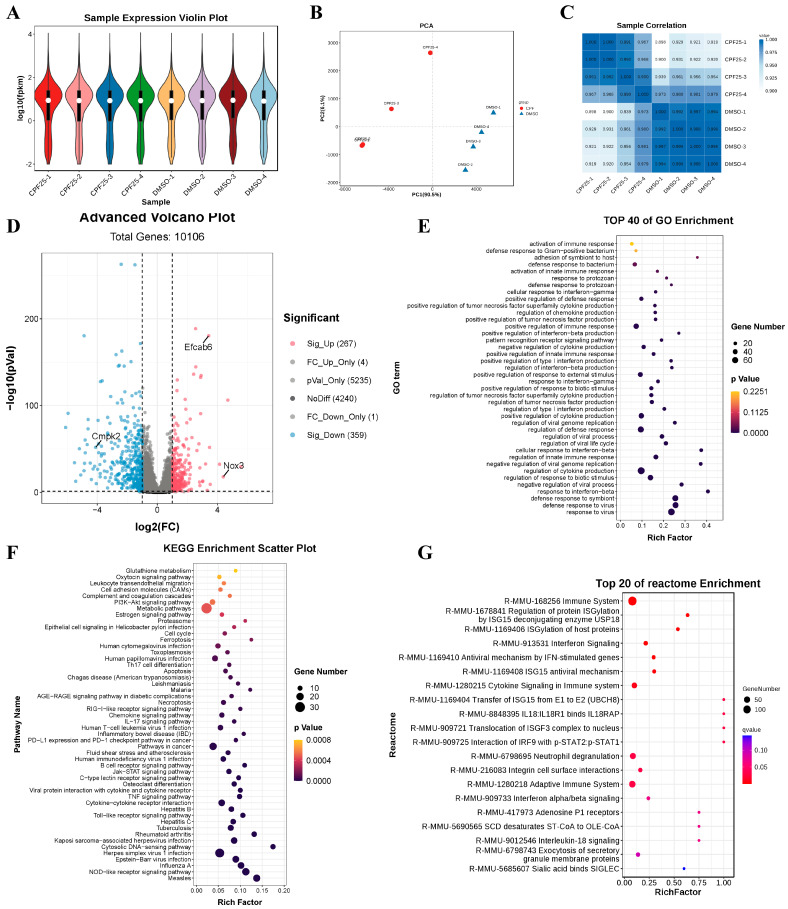
Analysis of RNA sequencing data for GC2*_spd_* cells after CPF treatment for 24 h. (**A**) Sample expression violin plot; (**B**) principal component analysis of sample; (**C**) sample correlation heat map; (**D**) statistical chart of differential genes; pink dots represent significantly up-regulated genes, blue dots represent significantly down-regulated genes, and the rest are non-significant genes. (**E**) GO enrichment classification histogram; (**F**) KEGG enrichment bubble plot; (**G**) Reactome enrichment bubble chart. Data presented are from four independent experiments (*n* = 4).

**Figure 6 cells-14-00940-f006:**
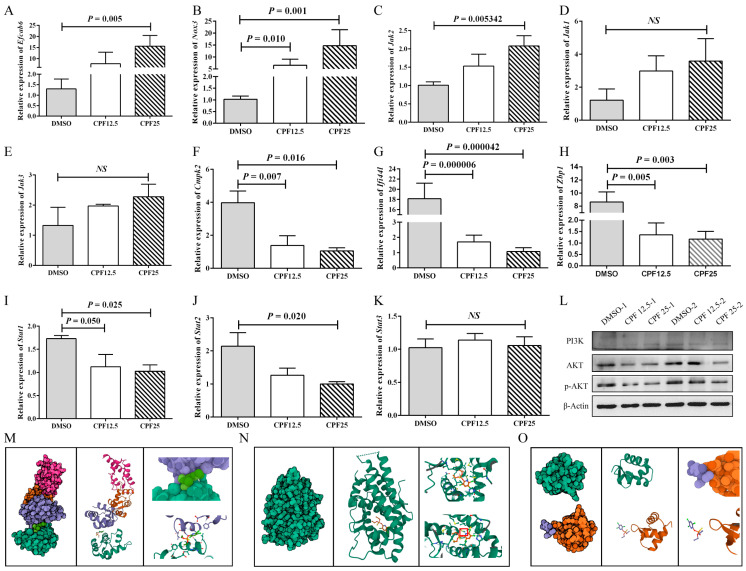
Validation of RNA sequencing results. (**A**) Expression of *Efcab6*; (**B**) expression of *Nox3*; (**C**) expression of *Jak2*; (**D**) expression of *Jak1*; (**E**) expression of *Jak3*; (**F**) expression of *Cmpk2*; (**G**) expression of *Ifi44l*; (**H**) expression of *Zbp1*; (**I**) expression of *Stat1*; (**J**) expression of *Stat2*; (**K**) expression of *Stat3*; (**L**) protein expression of PI3K-AKT pathway; (**M**) prediction of molecular docking of CPF and EFCAB6; (**N**) prediction of molecular docking of CPF and AR; (**O**) prediction of molecular docking of CPF and ZBP1. “*NS*” represents no significant difference between each group.

**Figure 7 cells-14-00940-f007:**
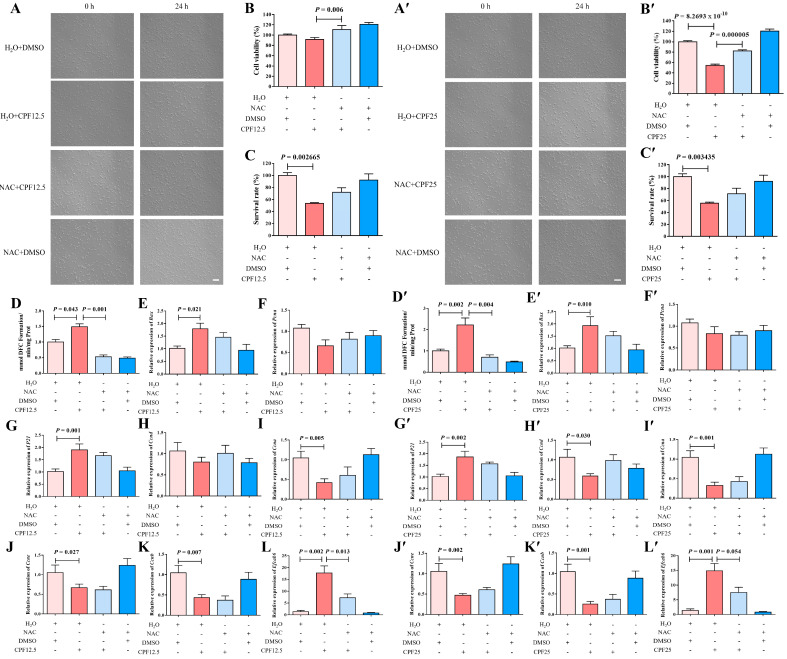
Restorative effect of NAC on GC2*_spg_* cells after CPF treatment for 24 h. (**A**–**L**) Restorative effect of NAC on GC2*_spg_* cells after 12.5 μmol/L CPF exposure; (**A**′–**L′**) restorative effect of NAC on GC2*_spg_* cells after 25 μmol/L CPF exposure; (**A**,**A′**) phenotypes of GC2*_spd_* cells; scale bar indicates 100 μm. (**B**,**B′**) cell viability of GC2*_spd_* cells; (**C**,**C′**) cell number of GC2*_spd_* cells; (**D**,**D′**) level of ROS in GC2*_spd_* cells; (**E**,**E′**) expression of *Bax*; (**F**,**F′**) expression of *Pcna*; (**G**,**G′**) expression of *P21*; (**H**,**H′**) expression of *Ccnd*; (**I**,**I′**) expression of *Ccna*; (**J**,**J′**) expression of *Ccne*; (**K**,**K′**) expression of *Ccnb*; (**L**,**L′**) expression of *Efcab6*. Scale bar indicates 100 μm.

**Table 1 cells-14-00940-t001:** The primers used for qRT-PCR.

Gene Names	Forward Primers (5′ to 3′)	Reverse Primers (5′ to 3′)	Production Sizes (bp)
*P21*	CCTGGTGATGTCCGACCTG	CCATGAGCGCATCGCAATC	103
*Ccnd*	TGCTGCAAATGGAACTGCTT	CCACAAAGGTCTGTGCATGCT	150
*Ccne*	GTGGCTCCGACCTTTCAGTC	CACAGTCTTGTCAATCTTGGCA	101
*Ccna*	GCCTTCACCATTCATGTGGAT	TTGCTGCGGGTAAAGAGACAG	118
*Ccnb*	AAGGTGCCTGTGTGTGAACC	GTCAGCCCCATCATCTGCG	228
*Efcab6*	CACCCTGAAAAGCAACACGG	TTAGCCTCCCCTTGGCATTG	131
*Cmpk2*	TTCTGAGGAGAGAGTGCGGA	AGATGGCAGCTTGGGTTCTC	138
*Zbp1*	AAGAGTCCCCTGCGATTATTTG	TCTGGATGGCGTTTGAATTGG	102
*Ifi44l*	GCTGTGTGATTCAATGGGGC	GCTCACAGGGGTTGAACTGA	116
*Igf2bp3*	GCTACGCGTTCGTGGACT	GTGGGGCGGGATATTTCGT	161
*Jak1*	GCCAGTGCCCTGAGTTACTT	TGTCTGGATCTTGCCTGGTC	166
*Jak2*	GGAATGGCCTGCCTTACAATG	TGGCTCTATCTGCTTCACAGAAT	108
*Jak3*	GATCTGCAAGGGCATGGAGT	ACTCCGGGGCATACCAAAAG	197
*Stat1*	TCACAGTGGTTCGAGCTTCAG	CGAGACATCATAGGCAGCGTG	150
*Stat2*	AGAAGTCCTGCATTGGAGCC	TTCAGTAGCTGCCGAAGGTG	106
*Vdac1*	GAGTATGGGCTGACGTTTACAG	GAGCTTCAGTCCACGAGCAAG	96
*Cox4i1*	ATTGGCAAGAGAGCCATTTCTAC	TGGGGAAAGCATAGTCTTCACT	82
*β-Actin*	CCTAAGGCCAACCGTGAAA	TGGTACGACCAGAGGCATA	112

## Data Availability

The original data of the paper are available upon request from the corresponding author.

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
