# Peer review of "CPF Induces GC2spd Cell Injury via ROS/AKT/Efcab6 Pathway"

_cells, 2025, doi:10.3390/cells14130940_

Round 1

Reviewer 1 Report

Comments and Suggestions for Authors

Comments about the manuscript:

“CPF induces GC2spd cell injury via ROS/AKT/Efcab6 pathway”

The massive use of pesticides has harmful repercussions on hulan and animal reproduction, particularly in males. The work presented seeks to understand the potential molecular mechanisms of the decline in male reproduction induced by chlorpyrifos (CPF), a particularly effective insecticide. To this end, the authors used several techniques (flow cytometry, qRT-PCR, Western Blot, RNA sequencing and bioinformatics analysis) to highlight the potential molecular mechanisms of damage caused to GC2spd cells cultured in vitro. The results showed that there were indeed harmful effects of this pesticide on the characteristics of cells cultured in the presence of the pesticide, in which several genes were then expressed differently compared to cells cultured without the pesticide. These results contribute to our understanding of the molecular mechanisms underlying CPF-induced male infertility.

I believe this work is important and could be published in Cells, albeit with some refinements to the manuscript. Here are a few comments.

Page 2, lines 86-87. “Based on this, the experiment employed CPF treatment on the GC2spd cell line (which leans towards spermatocytes)”. I think it is necessary to clarify some details about these cells: what is their embryonic origin, at what stage are they observed in the process of spermatogenesis? What is their position in the differentiation of spermatocytes and spermatozoa?

Page 3, line 112. “According to the manufacturer's instructions” does not seem sufficient to me for a scientific article; a brief description of the method seems desirable.

Page 3, line 112. “the viability of GC2spd”: I have not found the origin of this cell line, please provide some details (supplier, characteristics, etc.)

Page 3, line 119. “the live cells were collected by centrifugation.”: what were the characteristics of the centrifugation (speed in g number, duration).

Page 3, line 125. “according to the commercial instructions.” does not seem sufficient to me for a scientific article; a brief description of the method seems desirable.

Page 3, line 131. “The specific method was described in the kit instructions” does not seem sufficient to me for a scientific article; a brief description of the method seems desirable.

Page 3, line 141. “according to the manufacturer’s protocol” does not seem sufficient to me for a scientific article; a brief description of the method seems desirable.

Page 4, lines 149-150. “the fluorescent probe was added”: What was this fluorescent probe? Specify its characteristics and its preparation.

Page 5, line 153. “then centrifuged to collect the supernatant”: what were the characteristics of the centrifugation (speed in g number, duration).

Page 13, figure 7Figures 7A and 7A' are not clearly visible. They need to be improved (by increasing contrast or brightness, for example).

Page 14, line 451. “H2O+CPF12.5 group (Figure G, H, I)”: What is the number of the figure?

Supplementary file ; In Figure S2, figures A and B are not clearly visible; they require better contrast.

Author Response

Response to the reviewers’ comments on cells-3636441 B

Dear Editor and anonymous reviewers,

We greatly appreciate the anonymous reviewers for your careful review and constructive comments (cells-3636441 B). We have studied comments carefully and tried our best to revise the manuscript, and we hope that the revision can meet with your approval.

Comments from Reviewer 1 and author responses:

Comments 1: Page 2, lines 86-87. “Based on this, the experiment employed CPF treatment on the GC2spd cell line (which leans towards spermatocytes)”. I think it is necessary to clarify some details about these cells: what is their embryonic origin, at what stage are they observed in the process of spermatogenesis? What is their position in the differentiation of spermatocytes and spermatozoa?

Response 1: Thank you for your reminder. According to the description by ATCC and Li et al., GC2spd is a spermatocyte cell line (Li et al., 2024). Therefore, to avoid ambiguity, we will modify “Based on this, the experiment employed CPF treatment on the GC2spd cell line (which leans towards spermatocytes)” to “Based on this, the experiment employed CPF treatment on the GC2spd cell line (a spermatocytes cell line and purchased from ATCC, the Cat number was CRL-2196)”。

Reference

Li J, Chen D, Suo J, Li J, Zhang Y, Wang Y, Deng Z, Zhang Q, Ma B. Triptolide induced spermatogenesis dysfunction via ferroptosis activation by promoting K63-linked GPX4 polyubiquitination in spermatocytes. Chem Biol Interact. 2024. 399:111130.

Comments 2: Page 3, line 112. “According to the manufacturer's instructions” does not seem sufficient to me for a scientific article; a brief description of the method seems desirable.

Response 2: Thank you for your review. The specific methods have already been mentioned in the main text, but to enhance readability, we will modify this part to “According to the manufacturer's instructions, the viability of GC2spd cells was assessed after treatment with 0 and 25 μmol/L CPF for 12 hours using the Cell Counting Kit 8 (CCK8) (C0037, Beyotime Institute of Biotechnology, China). The specific experimental methods are as follows: 10 μL of CCK8 solution was added per 100 μL of cell culture medium; after incubation at 37 oC in darkness for 2~4 hours, the absorbance of the cells was measured at 450 nm using a microplate reader”.

Comments 3: Page 3, line 112. “the viability of GC2spd”: I have not found the origin of this cell line, please provide some details (supplier, characteristics, etc.)

Response 3: Thank you for your meticulous review. We have supplemented the introduction of the origin of the GC2spd cells in section 2.1 Cell culture and treatments. The details are as follows: The GC2spd cell line was purchased from ATCC: The Global Bioresource Center (https://www.atcc.org) (CRL-2196, ATCC, America). The medium contained DMEM/HIGH GLUCOSE (E600003, Sangon, China), fetal bovine serum (Z7186FBS-500, Zeta Life, America), Penicillin-Streptomycin (15140122, Gibco, America), GlutaMAX (35050061, Gibco, America), AdvanceSTEM ES Qualified Non-Essential Amino Acids (100X) (SH30853.01, HyClone, America), and Sodium Pyruvate (2276850, Gibco, America); their proportions were 86%, 10%, 1%, 1%, 1% and 1%, respectively.

Comments 4: Page 3, line 119. “the live cells were collected by centrifugation.”: what were the characteristics of the centrifugation (speed in g number, duration).

Response 4: Thank you for your valuable feedback. We have revised this section to “After treating GC2spd cells with 0 and 25 μmol/L CPF for 24 hours, the floating dead cells in the supernatant were removed, and the live cells digested with trypsin were collected by centrifugation at 450 g for 5 min”.

Comments 5: Page 3, line 125. “according to the commercial instructions.” does not seem sufficient to me for a scientific article; a brief description of the method seems desirable.

Response 5: Thank you for your careful review. The specific steps have already been described later in the text -- More than 1×104 cells were required in each tube. Subsequently, the treated cells were stained using Annexin-V-FLUOS and PI, respectively, and apoptosis was detected after incubation. The specific steps are described in Chen et al. [22].

Comments 6: Page 3, line 131. “The specific method was described in the kit instructions” does not seem sufficient to me for a scientific article; a brief description of the method seems desirable.

Response 6: Thank you for your feedback. We have added content to this section-- Firstly, GC2spd cells were fixed using 4 % paraformaldehyde at room temperature for 20-30 min, and the fixed samples were immersed in PBS and rinsed 3 times for 5 min each time; subsequently, the cells were permeabilised using 0.2% Triton-100 at 37 oC for 10 min; the permeabilised samples were immersed in PBS and rinsed 3 times for 5 min each time; 100 μL of TdT Equilibration Buffer was added to each sample, equilibrate at 37 oC for 30 min; prepare the labelling working solution and add an appropriate amount to each well, react at 37 oC for 60 min in dark; rinse the labelled samples by immersing them in PBS for 3 times, each time for 5 min; add DAPI working solution and incubate for 5 min at room temperature, avoiding light; rinse the samples by adding PBS for 4 times, each time for 5 min; Finally, the stained cells were photographed and counted under a fluorescence microscope to calculate the late apoptosis rate of GC2spd cells after 25 μmol/L CPF treatment.

Comments 7: Page 3, line 141. “according to the manufacturer’s protocol” does not seem sufficient to me for a scientific article; a brief description of the method seems desirable.

Response 7: Thank you for your suggestion. We have made the following revisions to this section: RNA in GC2spd cells treated with CPF (0, 12.5 and 25 μmol/L) was extracted using RNAiso Plus (9109, TAKARA, China), chloroform, isopropanol, anhydrous ethanol and enzyme-free water (R1600, Solarbio, China). The RNA purity of samples was assessed by its OD260/280 value using a NanoDrop 1000 spectrophotometer (Thermo Fisher Scientific Inc., Wilmington, DE, USA). In addition, electrophoretic analysis was performed on 1 % agarose gel to assess the RNA quality [24]. Then the cDNA of the GC2spd cells was obtained from RNA using Hifair® â…¡ 1st Strand cDNA Synthesis SuperMix for qPCR (gDNA digester plus) (11123ES10, YEASEN, China) according to the manufacturer’s protocol. Briefly, genomic DNA is first removed, followed by reverse transcription of RNA to cDNA. cDNA obtained was stored in a -80oC refrigerator. The gene expression levels in GC2spd cells were detected using qRT-PCR and performed using ChamQ SYBR qPCR Master Mix (Q311-02, vazyme, China). The primer sequences were designed by NCBI (https://blast.ncbi.nlm.nih.gov/Blast.cgi) and were shown in Table 1, and the specific qRT-PCR reaction system and reaction procedures were shown in Table S1 and Table S2.

Comments 8: Page 4, lines 149-150. “the fluorescent probe was added”: What was this fluorescent probe? Specify its characteristics and its preparation.

Response 8: Thank you for your careful reading. This section has been revised to: Briefly, the GC2spd cells were collected, the fluorescent probe DCFH-DA in the ROS assay kit was added, and bath in dark at 37 °C for 30 min.

Comments 9: Page 5, line 153. “then centrifuged to collect the supernatant”: what were the characteristics of the centrifugation (speed in g number, duration).

Response 9: Thank you for your thorough review. We have added the experimental conditions to this section. The revised content is as follows: Proteins were collected from GC2spd cells after CPF treatment for 24 hours using RIPA (PL001, Shaanxi Zhonghui Hecai Biological Medicine Technology Co., LTD, China), then centrifuged at 12000 g for 20 min in 4 oC to collect the supernatant, followed by addition of loading buff (P1040, Solarbio, China) and bath at 100 oC for 20 min.

Comments 10: Page 13, figure 7Figures 7A and 7A' are not clearly visible. They need to be improved (by increasing contrast or brightness, for example).

Response 10: Thank you for your suggestion. To address this issue, firstly, we adjusted the brightness and contrast of the images to facilitate your observation. Secondly, we visualized the number of cells in the images by creating bar charts (Figure 7C and Figure 7C’). Finally, we improved the clarity of the uploaded images to better present our results.

Comments 11: Page 14, line 451. “H2O+CPF12.5 group (Figure G, H, I)”: What is the number of the figure?

Response 11: Thank you for your thorough review. The images for this section are all in Figure 7. We have revised the figure numbering for this section and checked it throughout the entire manuscript.

Comments 12: Supplementary file ; In Figure S2, figures A and B are not clearly visible; they require better contrast.

Response 12: Thank you for your suggestion. Consistent with the approach for Question 10, we first adjusted the brightness and contrast of the images to enhance visibility. Next, we improved the clarity of the uploaded images to better present our results.

Reviewer 2 Report

Comments and Suggestions for Authors

In the work titled “CPF induces GC2spd cell injury via ROS/AKT/Efcab6 pathway” the authors aimed to explore the potential molecular mechanisms of male reproductive decline that induced by CPF. In particular, they through flow cytometry, qRT-PCR, Western Blot, RNA sequencing, and bioinformatics analysis investigate the potential molecular mechanisms of CPF-induced male reproductive damage in the GC2spd cells.

I find a very interesting work.  The work is well written and very clear but some information is missing. Therefore, I suggest a major revision.

 The suggestions I would make are as follows:

  • Better explain the choice of the dose used: 0 and 25 μmol/L CPF
  • Line 118: “After treating GC2spd cells with 0 and 25 μmol/L CPF for 24 hours, the floating dead”… define the percentage of dead cells
  • Better describe the following paragraph: 8 Western Blot (WB)
  • Add more details to this paragraph: 2.5 RNA extraction and qRT-PCR
  • Better describe the following paragraph: 2.6 The detection of oxidative stress level
  • Environmental pollution has become one of the main causes of infertility. I think the authors should say something about the effects of pollution on fertility. In this context, I better argue on the decline of semen quality in the last decades.
  • In order to amply the mechanism of oxidative damage and the possibility that also this contaminant can produce metabolomic alterations I suggest to read and quote papers in which these alterations are reported in various tissue including gonads and spermatozoa. Better link the results with each other for a better understanding of the results.
  • Better explain the action mechanism of NAC
  • Also the choice of the dose of NAC used must be explained
  • A molecular mechanism regarding the action of this contaminant must also be better hypothesized. I suggest to read and quote the following paper: 10.1016/j.cbpc.2023.109778
  • Better define the limitations of this study.
  • An accurate check for English language is extremely necessary in all the manuscript

Comments on the Quality of English Language

An accurate check for English language is extremely necessary in all the manuscript

Author Response

Response to the reviewers’ comments on cells-3636441 B

Dear Editor and anonymous reviewers,

We greatly appreciate the anonymous reviewers for your careful review and constructive comments (cells-3636441 B). We have studied comments carefully and tried our best to revise the manuscript, and we hope that the revision can meet with your approval.

Comments from Reviewer 2 and author responses:

Comments 1: Better explain the choice of the dose used: 0 and 25 μmol/L CPF

Response 1: Thank you for your valuable comments. The choice of 0 and 25 μmol/L CPF for the experiments was made for two main reasons. First, in our previously published study, mouse sperm were treated with a concentration of 25 μmol/L CPF. To ensure continuity in our experimental approach, we continued to use 25 μmol/L CPF to treat GC2spd cells (Zhang et al., 2020). Second, to compare the similarities and differences in CPF-induced male reproductive damage between mice and pigs, we used the same concentration as in our previous study (Zhang et al., 2023).

Reference

Zhang X, Cui W, Wang K, Chen R, Chen M, Lan K, Wei Y, Pan C, Lan X. Chlorpyrifos inhibits sperm maturation and induces a decrease in mouse male fertility. Environ Res. 2020. 188:109785.

Zhang X, Li M, Li W, Yue L, Zhang T, Tang Q, Zhang N, Lan X, Pan C. Chlorpyrifos induces male infertility in pigs through ROS and PI3K-AKT pathway. iScience. 2023. 26(5):106558.

Comments 2: Line 118: “After treating GC2spd cells with 0 and 25 μmol/L CPF for 24 hours, the floating dead”… define the percentage of dead cells

Response 2: Thank you for your review. The the percentage of dead cells has been accounted in Section 3.1 The morphology and viability of GC2spd cells after CPF treatment. It was observed that, the GC2spd cells after 24 hours of CPF treatment, the CPF12.5 and CPF25 treatment groups exhibited approximately a 20% decrease in cell number (Figure 1B).

Comments 3: Better describe the following paragraph: 8 Western Blot (WB)

Response 3: Thank you for your precious advice. We have revised section 2.8 as follows: Proteins were collected from GC2spd cells after CPF treatment for 24 hours using RIPA (PL001, Shaanxi Zhonghui Hecai Biological Medicine Technology Co., LTD, China), then centrifuged at 12000 g for 20 min in 4 oC to collect the supernatant, followed by addition of loading buff (P1040, Solarbio, China) and bath at 100 oC for 20 min. Subsequently, the prepared protein samples were subjected to electrophoresis, transferred to PVDF membranes (ISEQ00010, Millipore, America), seal with skimmed milk at room temperature for 2 h, and incubation with primary antibodies at 4 oC for 12 h, then secondary antibodies incubate at room temperature for 2 h after purged in TBST for 3 times. Protein expression was detected using Ultra-sensitive ECL Reagent (DY30208, DIYIBIO, China) and Bio-Rad Chemidoc. Details of primary and secondary antibodies were shown in Table S3.

Comments 4: Add more details to this paragraph: 2.5 RNA extraction and qRT-PCR

Response 4: Thank you for your comment. We have supplemented and revised the relevant section accordingly. The specific revisions are as follows: RNA in GC2spd cells treated with CPF (0, 12.5 and 25 μmol/L) was extracted using RNAiso Plus (9109, TAKARA, China), chloroform, isopropanol, anhydrous ethanol and enzyme-free water (R1600, Solarbio, China). The RNA purity of samples was assessed by its OD260/280 value using a NanoDrop 1000 spectrophotometer (Thermo Fisher Scientific Inc., Wilmington, DE, USA). In addition, electrophoretic analysis was performed on 1 % agarose gel to assess the RNA quality [24]. Then the cDNA of the GC2spd cells was obtained from RNA using Hifair® â…¡ 1st Strand cDNA Synthesis SuperMix for qPCR (gDNA digester plus) (11123ES10, YEASEN, China) according to the manufacturer’s protocol. Briefly, genomic DNA is first removed, followed by reverse transcription of RNA to cDNA. cDNA obtained was stored in a -80oC refrigerator. The gene expression levels in GC2spd cells were detected using qRT-PCR and performed using ChamQ SYBR qPCR Master Mix (Q311-02, vazyme, China). The primer sequences were designed by NCBI (https://blast.ncbi.nlm.nih.gov/Blast.cgi) and were shown in Table 1, and the specific qRT-PCR reaction system and reaction procedures were shown in Table S1 and Table S2.

Comments 5: Better describe the following paragraph: 2.6 The detection of oxidative stress level

Response 5: Thank you for your recommendation. We have provided a more detailed description of the detection of oxidative stress level to help readers better reproduce our experiments. The specific revisions can be found in Section 2.6 The detection of oxidative stress level.

Comments 6: Environmental pollution has become one of the main causes of infertility. I think the authors should say something about the effects of pollution on fertility. In this context, I better argue on the decline of semen quality in the last decades.

Response 6: Thank you very much for your valuable suggestions. We have added supplementary descriptions regarding the decline in semen quality in the Introduction section. The detailed revisions can be found in the manuscript.

Comments 7: In order to amply the mechanism of oxidative damage and the possibility that also this contaminant can produce metabolomic alterations I suggest to read and quote papers in which these alterations are reported in various tissue including gonads and spermatozoa. Better link the results with each other for a better understanding of the results.

Response 7: Thank you for your careful review of the manuscript, which we have briefly described in the Introduction section on oxidative stress and its relationship to male reproduction.

Comments 8: Better explain the action mechanism of NAC

Response 8: The mechanisms by which NAC (N-acetylcysteine) scavenges ROS are primarily as follows (Kalyanaraman, 2022):

  1. NAC is an acetylated derivative of cysteine. Once it enters the cell, it is metabolized into cysteine, which is then used to synthesize glutathione (GSH). GSH is one of the most important intracellular antioxidants. It can reduce the accumulation of ROS by catalyzing the conversion of hydrogen peroxide (Hâ‚‚Oâ‚‚) and lipid peroxides into harmless water and alcohols through reactions mediated by glutathione peroxidase (GPx).
  2. NAC can also modulate the intracellular redox state, thereby affecting the activity of antioxidant enzymes and further inhibiting the generation of ROS.

We have also described these mechanisms in our manuscript: NAC scavenges intracellular ROS by enhancing the intracellular cysteine pool, increasing glutathione (GSH) levels and enhancing the activity of antioxidant enzymes (glutathione peroxidase, thioredoxin) [25].

Reference

Kalyanaraman B. NAC, NAC, Knockin' on Heaven's door: Interpreting the mechanism of action of N-acetylcysteine in tumor and immune cells. Redox Biol. 2022. 57:102497.

Comments 9: Also the choice of the dose of NAC used must be explained

Response 9: According to the published article, it has been demonstrated that in GC2spd cells, a concentration of 5 mmol/L NAC enhances cell viability and exhibits a pronounced restorative effect on cell damage induced by MeHg (Zhang et al., 2023). Similarly, Wei et al. reported that 5 mmol/L NAC effectively attenuated BDE-209-induced oxidative stress in GC2spd cells (Wei et al., 2023). In addition, our experiments also confirmed that 5 mmol/L NAC is non-toxic to GC2spd cells. Finally,We have also addressed the selection of the NAC dose in the article.

Reference

Zhang X, Hao H, Ma K, Pang H, Li X, Tian T, Hou S, Ning X, Wu H, Hou Q, Li M, Sun Y, Song X, Jin M. The role and mechanism of unfolded protein response signaling pathway in methylmercury-induced apoptosis of mouse spermatocytes germ cell-2 cells. Environ Toxicol. 2023. 38(2):472-482.

Wei Y, Geng W, Zhang T, He H, Zhai J. N-acetylcysteine rescues meiotic arrest during spermatogenesis in mice exposed to BDE-209. Environ Sci Pollut Res Int. 2023. 30(17):50952-50968.

Comments 10: A molecular mechanism regarding the action of this contaminant must also be better hypothesized. I suggest to read and quote the following paper: 10.1016/j.cbpc.2023.109778

Response 10: We have cited this article in our manuscript, which can be seen in section 2.5 RNA extraction and qRT-PCR.

Comments 11: Better define the limitations of this study.

Response 11: Thank you for your review, we have outlined the limitations of the study and subsequent experiments that need to be performed in the last part of the discussion. Specifically: In this study, we examined in detail the potential mechanisms of male injury induced by CPF. However, unfortunately, we have not yet performed recovery experiments in vivo using NAC. In subsequent studies we will aim to further investigate the restorative effects of NAC on CPF-induced male reproductive injury in vivo and to find natural compounds to replace the function of NAC from a food-borne perspective.

Comments 12: An accurate check for English language is extremely necessary in all the manuscript

Response 12: Thank you for your suggestions. The corresponding author of the article, Prof Chuan-Ying Pan, was a visiting scholar in United States for many years. We have revised the language of the article in detail before submission (as shown below). During this revision period, we have also revised the language of the article in order to meet the publication requirements of the journal.

Round 2

Reviewer 2 Report

Comments and Suggestions for Authors

The authors addressed all my questions. I accept the manuscript in the present form